# Discriminatively Unsupervised Learning Person Re-Identification via Considering Complicated Images

**DOI:** 10.3390/s23063259

**Published:** 2023-03-20

**Authors:** Rong Quan, Biaoyi Xu, Dong Liang

**Affiliations:** School of Computer Science and Technology, Nanjing University of Aeronautics and Astronautics, Nanjing 210016, China; rongquan0806@gmail.com (R.Q.); xubiaoyi@nuaa.edu.cn (B.X.)

**Keywords:** purely unsupervised learning person re-ID, contrastive loss, unclustered outliers, complicated images

## Abstract

State-of-the-art purely unsupervised learning person re-ID methods first cluster all the images into multiple clusters and assign each clustered image a pseudo label based on the cluster result. Then, they construct a memory dictionary that stores all the clustered images, and subsequently train the feature extraction network based on this dictionary. All these methods directly discard the unclustered outliers in the clustering process and train the network only based on the clustered images. The unclustered outliers are complicated images containing different clothes and poses, with low resolution, severe occlusion, and so on, which are common in real-world applications. Therefore, models trained only on clustered images will be less robust and unable to handle complicated images. We construct a memory dictionary that considers complicated images consisting of both clustered and unclustered images, and design a corresponding contrastive loss by considering both kinds of images. The experimental results show that our memory dictionary that considers complicated images and contrastive loss can improve the person re-ID performance, which demonstrates the effectiveness of considering unclustered complicated images in unsupervised person re-ID.

## 1. Introduction

Purely unsupervised learning person re-identification (re-ID) refers to recognizing the same person appearing in images captured by different cameras without any groundtruth labels. With the rapid development of unsupervised representation learning [1,2,3], the performance of purely unsupervised learning person re-ID based on contrastive loss has been gradually improving [4,5,6,7,8]. On the other hand, purely unsupervised learning person re-ID does not require any labeled information or complex training processes, which is easy to implement and deploy in real-world application scenarios. Therefore, purely unsupervised learning person re-ID has great research potential and a promising future.

Most existing purely unsupervised learning person re-ID methods execute iterative training in the following way. Firstly, they generate an initial feature representation for each image from the initial feature extraction network. Based on the initial image feature representations, they cluster all the images into multiple clusters and assign a pseudo label to each clustered image according to the cluster results. Knowing the image feature representations and pseudo labels, these methods build an instance-level [9,10] or cluster-level [11] memory dictionary. Then, they sample a batch of query images and compute the contrastive loss by comparing the query images with all the instance representations in the memory dictionary. Next, these methods train the feature extraction network based on the contrastive loss, and update the instance representations in the memory dictionary based on the trained feature extraction network.

Memory dictionary construction and contrastive loss calculations are two key components of existing purely unsupervised learning person re-ID methods. Ge et al. [9] constructed an instance-level memory dictionary consisting of all the clustered images and computed a cluster-level contrastive loss. They first generated each cluster’s representation by averaging the features of all the images in this cluster, and then computed the cluster-level contrastive loss by comparing the query samples with all the cluster representations. Based on the obtained contrastive loss, they updated the feature representations of the query images’ corresponding instances in the dictionary. Using the cluster-level contrastive loss could reduce the error caused by the noisy pseudo labels and also simplify the calculation. However, the instance-level feature update in the memory dictionary may lead to an unbalanced update and inconsistent representation for the clusters, especially when the numbers of images contained in each cluster varies significantly [11]. To solve this problem, Dai et al. [11] directly constructed a cluster-level memory dictionary and used the cluster centroid as each cluster’s representation in the dictionary. They computed the cluster-level contrastive loss by comparing the query images with all the cluster representations, and directly updated the cluster representations based on the query images. Computing the contrastive loss and updating the feature representations in the cluster-level can avoid the unbalanced cluster update and inconsistent cluster representation problems, and thus result in a better performance. We follow [11] and use the cluster-level memory dictionary and contrastive loss in our work.

We found that the existing cluster-based purely unsupervised learning person re-ID methods directly discard the unclustered outliers in the clustering process and train the feature extraction network only based on the clustered images. Through our observations, these unclustered outliers often contain pedestrians that are difficult to recognize owing to the problem of large variations in clothes and pose, severe occlusion, low resolution, different light, and so on. Such kinds of complicated images are also frequently encountered, especially in real-world applications. Directly ignoring these complicated images and training the feature extraction network only based on easily clustered common images will result in a less robust trained network which is unable to handle complicated images. Therefore, other than the easily clustered common images, we also consider complicated images that are hard to cluster during the clustering process in our method.

We propose an unsupervised learning person re-ID method that considers complicated images based on both easily clustered common images and unclustered complicated images in our work. Specifically, we propose a novel memory dictionary that considers complicated images and stores both clustered common images and unclustered complicated images, and a corresponding cluster-level contrastive loss which compares the query instances with not only the positive and negative instances, but also the unclustered complicated instances in the dictionary. We make an assumption that the unclustered complicated images always have different person IDs with the query images, and are difficult to recognize. Based on this assumption, we construct our cluster-level dictionary as follows. For the clustered images, we average the feature representations of all the images in each cluster and use the obtained average feature as the cluster’s feature representation. For the unclustered complicated images, we randomly sample one image from the complicated images and use its feature to represent the unclustered complicated instance in the dictionary. When computing the contrastive loss, we consider three types of instances in the dictionary for each query image, i.e., a positive instance which contains the same person as the query image, a negative instance of the opposite situation, and the complicated instance. Since we assume that the complicated instance has a different person in the query images and is difficult to distinguish, we treat the complicated images differently during the calculation of the contrastive loss.

The main contributions of this paper are three-fold:We exploit the unclustered complicated images in the clustering stage to increase the trained model’s ability to recognize various images, and thus make our method more robust and suitable for real-world complex applications.We construct a novel memory dictionary which considers complicated images, consisting of both easily clustered common images and unclustered complicated images, and design a more effective contrastive loss by comparing the query samples with not only the positive and negative instances, but also the complicated instances in the dictionary.We demonstrate that our proposed method outperforms other state-of-the-art purely unsupervised learning person re-ID methods and some unsupervised domain adaptation methods.

## 2. Related Works

Existing unsupervised person re-ID methods can be divided into two classes: unsupervised domain adaptation methods and purely unsupervised learning person re-ID methods. Unsupervised domain adaptation methods use information from additional source domains to help train the model and update the feature representations in the target domain, while the purely unsupervised learning methods train the model completely based on unlabeled data.

*Unsupervised domain adaptation.* Unsupervised domain adaptation methods utilize transfer learning to improve the person re-ID performance on the target domain. Existing unsupervised domain adaptation methods can further be divided into two main categories: pseudo label-based methods [4,12,13,14,15,16,17,18] and domain translation-based methods [19,20,21,22,23]. Pseudo label-based methods first pre-train the model on the source domain and extract the features of the instances in the target domain based on the pre-trained model. Next, they generate pseudo labels for the instances in the target domain by either clustering their features or by measuring their feature similarities with the example features. Clustering-based methods are the most common unsupervised domain adaption person re-ID methods since they achieve a state-of-the-art performance. Two key characteristics of clustering-based methods are generating pseudo labels with higher accuracies and preventing the final person re-ID results from suffering from the noisy pseudo labels. Toward this end, Fu et al. [4] explored both the whole body and local body parts similarity to construct multiple clusters, and thus generate more accurate multi-scale pseudo labels. Ge et al. [13] proposed a mutual mean-teaching method to alleviate the errors induced by noisy pseudo labels which are generated from directly clustering on the target domain. They gradually refined the pseudo labels in the target domain by alternatively refining the hard pseudo labels offline and the soft pseudo labels online. Zhai et al. [12] proposed an augmented discriminative clustering method to more extensively use rich unlabeled images in the target domain. re-ID models based on augmented clusters are more discriminative. Zhang et al. [12] proposed an augmentation framework-based self-training method, which progressively improved the model’s performance by alternatively executing conservative and promising stages. Although unsupervised domain adaptation re-ID methods can achieve a promising performance by exploiting information from the source domain, they still need supervised information on the source domain and their performances are susceptible to the difference between the target domain and the source domain.

*Purely unsupervised learning person re-ID.* Purely unsupervised learning person re-ID methods completely train models on unlabeled data [5,6,7,9,10,24,25,26,27,28]. The training process usually consists of four main steps, including clustering to generate pseudo labels, constructing a memory dictionary, computing the contrastive loss, and updating the feature representations. Pseudo label generation and memory dictionary construction are two keys parts of this kind of method. Lin et al. [5] proposed a bottom-up clustering method by considering both the diversity between images with different people and the similarity between images with the same person. They treated each image as an individual cluster and gradually grouped similar images into one cluster to finally obtain a great balance between the diversity across clusters and similarity in clusters. Wang et al. [6] exploited camera-aware proxies in each cluster to further distinguish the images from different cameras, and thus generate more reliable pseudo labels. Based on the camera-aware pseudo labels, they designed both intra-camera and inter-camera contrastive loss to enhance the model’s identity discrimination ability. Wang et al. [10] formulated person re-ID as a multi-label classification problem. They first assigned a single-class image label and then proposed a memory-based multi-label classification loss to merge the single-label and multi-label classification into one framework. Ge et al. [9] proposed a self-paced contrastive learning method to gradually generate more accurate cluster results. Based on this, they calculated more accurate feature representations for the memory dictionary. Zhang et al. [27] refined the pseudo labels temporally based on the pseudo label similarities between every two successive training iterations with clustering consensus. The current purely unsupervised learning person re-ID methods leverage various means to remove the label noise and obtain more accurate pseudo labels, and update the feature representations based on these refined pseudo labels. However, these methods directly discard unclustered outliers when generating pseudo labels, which are complicated images that are hard to cluster owing to large variations in clothes and pose, severe occlusion, low resolution, different light, and so on. Training the model solely based on the easily clustered images will result in the model being unable to handle complicated images and impractical for real-world scenarios. In this work, we construct a memory dictionary and train the feature extraction network based on both easily clustered images and unclustered complicated images to increase the trained model’s generalization ability.

## 3. Methods

Figure 1 shows the overall framework of the proposed purely unsupervised learning person re-ID method considering complicated images. Given a set of *N* training images $\left\{I_{1},I_{2},\ldots ,I_{N}\right\}$, we first use the feature extraction network $f_{\theta }$ to generate the initial image feature representations $F=\{f_{1},f_{2},\ldots ,f_{N}\}$, and cluster the image feature representations into *K* clusters $\left\{C_{1},C_{2},\ldots ,C_{K}\right\}$. Next, we construct a cluster-level memory dictionary *D* with K+1 items, including *K* cluster centroids $\left\{c_{1},c_{2},\ldots ,c_{K}\right\}$ and the image feature representation h of a random unclustered complicated image. Next, we sample a batch of *M* query images $\left\{Q_{1},Q_{2},\ldots ,Q_{M}\right\}$ from the training dataset, input them into the feature extraction network to obtain the query feature representations $\left\{q_{1},q_{2},\ldots ,q_{M}\right\}$, and compute a cluster-level contrastive loss by comparing the query representations with the positive, negative, and complicated feature representations in *D*. At last, we train the feature extraction network with this contrastive loss and use the trained network to update the cluster centroids in *D*. The blue arrows in Figure 2 show the network training process and the pink arrows show the representative updating process of *D* after network training. Figure 2 illustrates one iteration of the method training, and it will take many iterations to achieve the optimal person re-ID result. Next, we will introduce the details of the proposed method.

### 3.1. Memory Dictionary Construction

Existing purely unsupervised learning person re-ID methods first cluster all the training images into multiple clusters and assign each clustered image a pseudo label based on the cluster results. They directly discard the unclustered outliers and completely ignore them during network training and feature updating processes, which is unreasonable since the unclustered images are also very important. These unclustered complicated images mostly contain pedestrians with severe occlusion, low resolution, or large variations in clothes, pose, or light, which are frequently present in real-world application scenarios. Such kinds of complicated images are the focus of most supervised learning person re-ID methods, but are discarded and completely ignored by most purely unsupervised learning person re-ID methods. Consequently, the trained network will fail to handle complicated images and thus have low robustness and generalization ability.

To take all the images into consideration during training, we construct a novel memory dictionary which considers complicated images, consisting of both easily clustered common images and the unclustered complicated images. Specifically, we first use a widely used cluster method named DBSCAN [29] to cluster all the images into *K* clusters $\left\{C_{1},C_{2},\ldots ,C_{K}\right\}$. For each cluster $C_{i}$, we compute its cluster centroid $c_{i}$ as the average feature of all the images belonging to it:(1)$$c_{i}=\frac{1}{|C_{i}|}\sum_{j=1}^{|C_{i}|}f_{j}$$
where $|C_{i}|$ represents the number of the images in cluster $C_{i}$ and $f_{j}$ represents the feature of the *j*th image in cluster $C_{i}$.

For the unclustered images in the clustering process, we assume that they are different from the query images, while very difficult to recognize. Considering that the unclustered images are also different from each other and may contain images of the same person as the query image, we do not simply use all the unclustered images or their average features to represent the complicated image set in the dictionary. Instead, we randomly select one image from the unclustered images at each training iteration and use the selected image’s featured to represent the complicated image set of *D*. Although the unclustered images may contain images of the same person as the query images, randomly selecting one image at each training iteration can dramatically decrease the probability that the complicated image and the query image contain the same person.

We use labels 1 to K as the pseudo labels of all the clusters, and label 0 as the pseudo label of the complicated image set. After these operations, we construct a memory dictionary that considers complicated images, *D*, which covers all kinds of training images.

### 3.2. Contrastive Loss Calculation

After constructing the memory dictionary *D*, we randomly sample a batch of M=P×T query images from the training images. In detail, we sample images from person *P*, and sample *T* images for each person following the set in [11,30]. We design our contrastive loss which considers complicated images based on the ClusterNCE loss of [11]. Specifically, for each query image $Q_{i}$ with feature representation $q_{i}$, we compute its contrastive loss as:(2)$$L_{h}=-\log\frac{\exp\left(q_{i}\cdot c_{+}/\tau \right)}{\sum_{k=1}^{K}\exp\left(q_{i}\cdot c_{k}/\tau \right)+\eta \exp\left(q_{i}\cdot h/\tau \right)}$$
where $c_{+}$ is the representation of the positive cluster which $Q_{i}$ belongs to, $c_{k}$ is the representation of cluster $C_{k}$, h is the representation of the complicated images in the dictionary, τ is the temperature hyper-parameter, and η is the hyper-parameter used to balance the influences of the common and complicated images. This contrastive loss considering complicated images not only requires the query image representation to have the smallest distance to the positive cluster representation and the largest distances to the negative cluster representations, but also requires the distance between the query image representation and the complicated image representation to be the largest.

### 3.3. Training and Updating

After the final training iteration, the obtained contrastive loss is used to train the feature extraction network. By considering both the easily clustered common images and unclustered complicated images, the trained feature extraction network is able to generate more accurate feature representations for all the training images. Next, we update the cluster representations and unclustered image’s feature representation using the trained feature extraction network. We first use a momentum updating method [11] to update the cluster representation in the memory dictionary. Specifically, we use the trained feature extraction network to recalculate each query image’s feature representation, and based on this we update the representations of the clusters that the query images belong to. The updating formula is as follows:(3)$$c_{k}\leftarrow \mu c_{k}+\left(1-\mu \right)q_{i}^{*}$$
where $q_{i}^{*}$ is the recalculated feature representation of the query image $Q_{i}$ from the trained feature extraction network and $Q_{i}$ belongs to cluster $C_{k}$. $c_{k}$ is the original vector representation of cluster $C_{k}$ in *D* and μ is the momentum parameter that controls the updating degree of the cluster representation. A small μ value indicates a large change to the original cluster representation at each update, and a big μ value indicates a small change to the original cluster representation at each update. For the unclustered complicated images, we recalculate the feature representation of the selected complicated image and replace it in the complicated image set. In the next iteration, we again randomly select a complicated image’s feature representation as the representation of the complicated image set in *D*.

The above process describes one iterative training, and the whole training framework of our method is shown in Algorithm 1.
**Algorithm 1:** Training framework of the unsupervised learning person re-ID method which considers complicated images.
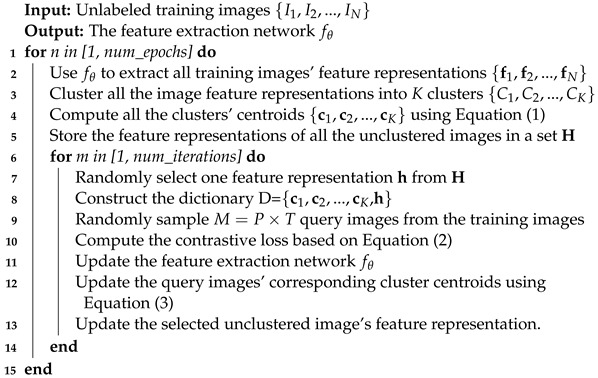


## 4. Results

### 4.1. Experimental Settings

#### 4.1.1. Datasets and Evaluation Metrics

To evaluate our proposed unsupervised learning person re-ID method which considers complicated images, we conducted experiments on two widely used person re-ID datasets named Market-1501 [31] and MSMT17 [19]. The Market-1501 dataset contains 32,668 images with 1501 distinct identities, which were collected from six cameras. The MSMT17 dataset contains 126,441 bounding images of 4101 distinct identities, which were collected from 15 cameras. The MSMT17 dataset is the most challenging and biggest person re-ID dataset, the images of which have signifiant scene and lighting variations. We use the mean average precision (mAP) and cumulative matching characteristic (CMC) top-1, top-5, and top-10 as the evaluation metrics of person re-ID.

#### 4.1.2. Implementation Details

We use ReNet-50 [32] as the backbone of the feature extraction network and parameters pretrained on ImageNet to initialize the network. Following the settings in [11], we remove all the sub-module layers after the fourth convolutional layer, add a global average pooling layer, batch normalization layer, and L2-normalization layer in turn, and finally output a 2048-dimensional feature representation for each image. Each image is resized into 256 × 128 pixels before it is input into the feature extraction network. Before training, we conduct data augmentation on each dataset by random horizontal flipping, cropping, and erasure. We set *P* = 16 and *T* = 16, i.e., we sample a batch of 256 images from 16 identities in each training iteration. We use the Adam optimizer to train the network parameters. We set τ to 0.01, μ to 0.1, and η to 2. The initial learning rate is 0.0035, and it is reduced to 1/20 of its original value every 20 epochs. We train the model for 80 epochs.

### 4.2. Comparison with Other Unsupervised Person Re-ID Methods

We compared our method with some other state-of-the-art unsupervised person re-ID methods on the Market-1501 and MSMT17 datasets, including both purely unsupervised learning (BUC [5], SSL [33], MMCL [10], HCT [34], CycAs [35], UGA [36], SPCL [9], IICS [7], OPLG [28], RLCC [27], ICE [25], PPLR [37], Cluster-ReID [11], TAUDL [38], and UTAL [39]) and some unsupervised domain adaptation methods (MMCL [10], AD-Cluster [12], MMT [13], SPCL [9], TDR [40], and ECN [41]). The comparison results are shown in Table 1 and Table 2, where the baselines with “*” are the unsupervised domain adaption person re-ID methods, and those without “*’; are purely unsupervised learning person re-ID methods. We can see from Table 1 and Table 2 that our method outperforms all the compared purely unsupervised learning person re-ID methods on the Market-1501 dataset, and is comparable with the state-of-the-art purely unsupervised learning person re-ID methods on the MSMT17 dataset. In addition, although the unsupervised domain adaptation methods exploit additional source domain information such as other labeled data or trained person re-ID models during their training process, our method still outperforms some state-of-the-art unsupervised domain adaptation methods, which further demonstrates the superiority of our proposed method. Our model is based on Cluster-ReID [11], where we further consider the unclustered outliers in the clustering process by designing a new memory dictionary that considers complicated images and contrastive loss. We can see from the last two rows of Table 1 and Table 2 that the performance of our method is better than Cluster-ReID. By comparing the performances of our method and Cluster-ReID, we can conclude that considering both the easily clustered common images and the unclustered complicated images in cluster-based purely unsupervised learning person re-ID can obtain better person re-ID performance than when only considering easily clustered common images.

### 4.3. Ablation Study

We use the unclustered outliers in the clustering process to help construct a memory dictionary considering complicated images in our method. We assume that the complicated images contain different people than the query samples and are difficult to recognize. Furthermore, considering that the unclustered complicated images are very likely to be different from each other, we randomly sample one image from the complicated images and use its features to represent the complicated instance in the memory dictionary. Only using one image of the complicated image set to represent the complicated instance at each training iteration can avoid the sampled complicated image containing the same person as the query image, which is inconsistent with our assumption and will cause calculation errors. To demonstrate the effectiveness of the sampling strategy, we conducted an ablation experiment to randomly sample more than one image from the complicated image set from the Market-1501 dataset. The experimental results are shown in Table 3, where the numbers in the first column represent the number of images sampled from the complicated image set in each training iteration. As we can see from Table 3, sampling more images from the complicated image set will reduce the person re-ID performance of our method, which demonstrates the effectiveness of sampling only one image to represent the complicated image set in the dictionary.

As mentioned before, we use the momentum updating method to update the vector representation of each cluster in the memory dictionary. The updating formula is shown in Equation (Equation 3). At each iteration and update, the updated vector representation of a cluster is composed of its original vector representation and the recalculated feature representations of the query images belonging to this cluster, where the ratios of the original vector representation and the recalculated image feature representations are μ and 1−μ, respectively. μ is known as the momentum in the momentum updating method. Usually, a small μ value indicates a significant change to the original cluster representation at each update, and a big μ value indicates a small change to the original cluster representation at each update. Here, we attempt to use different momentum values and observe the person re-ID performances when using these different momentum values. Specifically, we experiment with 33 different momentum values and their person re-ID performances are reported in Figure 2. As we can see from Figure 2, momentum values smaller than 0.9 can always generate good person re-ID performances, while momentum values larger than 0.9 always result in a poor performance. Therefore, we set μ as 0.1 in our method.

## 5. Conclusions

We proposed a novel cluster-based, purely unsupervised learning person re-ID method that considers complicated images, where we constructed a memory dictionary considering complicated images and contrastive loss by not only considering the easily clustered common images, but also the complicated images that are hard to cluster. At each iteration, we use the average features to represent each cluster instance, randomly sample a complicated image to represent the complicated instance in the memory dictionary, and compute the contrastive loss based on a comparison between the query images with both kinds of instances. The experimental results show that using our proposed memory dictionary and contrastive loss can clearly improve the person re-ID performance, which demonstrates the effectiveness of considering complicated images during the iterative training process, as well as the effectiveness of our method. We found that the sampled complicated image sometimes contains the same person as the query images, which is contrary to our assumption and will therefore introduce error into the final person re-ID result. To tackle this issue, we prefer to use a more effective complicated instance sample strategy to select a more useful complicated image at each iteration that definitely contains a different person compared to the query images.

## Figures and Tables

**Figure 1 sensors-23-03259-f001:**
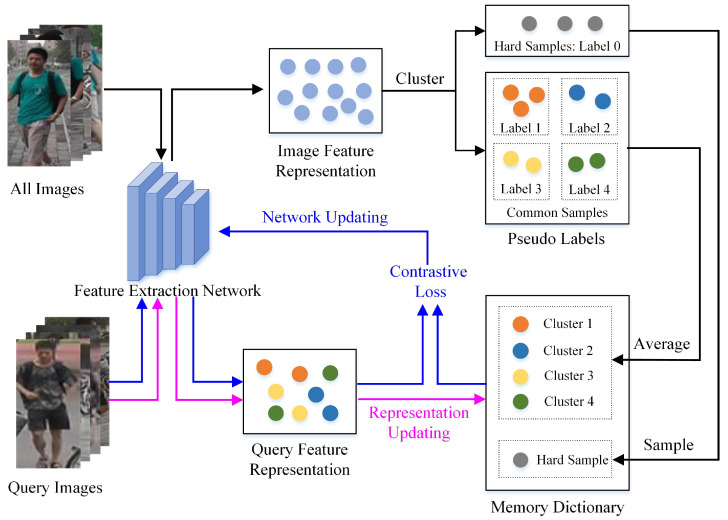
The flowchart of our proposed person re-ID method. The memory dictionary considering complicated images is made up of the clustered common images and unclustered complicated images, where the cluster instance is represented by the cluster centroid and the unclustered instance is represented by a randomly sampled complicated image. Furthermore, the contrastive loss is calculated by comparing the query images with two kinds of instances in the dictionary. The blue arrows show the training process of the feature extraction network and the pink arrows show the feature representation updating process.

**Figure 2 sensors-23-03259-f002:**
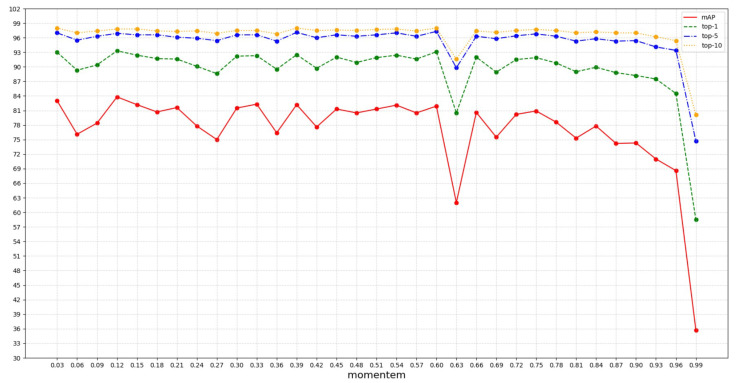
Performance comparison of using different momentum values during the momentum updating process. The horizontal axis represents the momentum values and the vertical axis represents the person re-ID performance.

**Table 1 sensors-23-03259-t001:** Comparison between our method and other state-of-the-art unsupervised person re-ID methods on the Market-1501 dataset. The methods marked with “*” are unsupervised domain adaptation person re-ID methods, while those without “*” are purely unsupervised learning person re-ID methods.

Method	Market-1501				
	Source	mAP	Top 1	Top 5	Top 10
MMCL [10] *	Duke	60.4	84.4	92.8	95.0
AD-Cluster [12] *	Duke	68.3	86.7	94.4	96.5
MMT [13] *	MSMT17	75.6	89.3	95.8	97.5
SPCL [9] *	MSMT17	77.5	89.7	96.1	97.6
TDR [40] *	Duke	83.4	94.2	-	-
BUC [5]	None	38.3	66.2	79.6	84.5
SSL [33]	None	37.8	71.7	83.8	87.4
MMCL [10]	None	45.5	80.3	89.4	92.3
HCT [34]	None	56.4	80.0	91.6	95.2
CycAs [35]	None	64.8	84.8	-	-
UGA [36]	None	70.3	87.2	-	-
SPCL [9]	None	73.1	88.1	95.1	97.0
IICS [7]	None	72.1	88.8	95.3	96.9
OPLG [28]	None	78.1	91.1	96.4	97.7
RLCC [27]	None	77.7	90.8	96.3	97.5
ICE [25]	None	79.5	92.0	97.0	98.1
PPLR [37]	None	81.5	92.8	97.1	98.1
Cluster-ReID [11]	None	83.0	92.9	97.2	98.0
Ours	None	83.8	93.3	96.9	97.8

**Table 2 sensors-23-03259-t002:** Comparison between our method and other state-of-the-art unsupervised person re-ID methods on the MSMT17 dataset. The methods with ‘*’ are unsupervised domain adaptation person re-ID methods, while those without ‘*’ are purely unsupervised learning person re-ID methods.

Method	MSMT17				
	Source	mAP	Top-1	Top-5	top-10
MMT [13] *	Market	24.0	50.1	63.5	69.3
SPCL [9] *	Market	26.8	53.7	65.0	69.8
ECN [41] *	Duke	10.2	30.2	41.5	46.8
MMCL [10] *	Duke	16.2	43.6	54.3	58.9
TDR [40] *	Duke	36.3	66.6	-	-
MMCL [10]	None	11.2	35.4	44.8	49.8
CycAs [35]	None	26.7	50.1	-	-
UGA [36]	None	21.7	49.5	-	-
SPCL [9]	None	19.1	42.3	55.6	61.2
TAUDL [38]	None	12.5	28.4	-	-
UTAL [39]	None	13.1	31.4	-	-
IICS [7]	None	18.6	45.7	57.7	62.8
OPLG [28]	None	26.9	53.7	65.3	70.2
RLCC [27]	None	27.9	56.5	68.4	73.1
ICE [25]	None	29.8	59.0	71.7	77.0
PPLR [37]	None	31.4	61.1	73.4	77.8
Cluster-ReID [11]	None	33.0	62.0	71.8	76.7
Ours	None	34.9	61.9	72.7	77.0

**Table 3 sensors-23-03259-t003:** Comparison of randomly sampling different numbers of images from the complicated image set on the Market-1501 dataset.

Number	Market-1501			
	mAP	Top 1	Top 5	Top 10
1	83.8	93.3	96.9	97.8
2	82.9	92.7	96.7	98.0
3	82.3	92.5	96.6	97.8
4	80.9	91.7	96.6	97.7

## Data Availability

Not applicable.

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
