# Peer review of "Discriminatively Unsupervised Learning Person Re-Identification via Considering Complicated Images"

_sensors, 2023, doi:10.3390/s23063259_

Round 1

Reviewer 1 Report

This paper presents an unsupervised person re-identification method that could learn from difficult samples e.g., persons with different clothes and low-quality images. To do so, the authors have proposed a memory dictionary and designed a contrastive loss to consider easy and difficult samples. The methodology is explained well, with a diagram that illustrates the overall framework and iterative training process. The experimental results demonstrate the effectiveness of the idea in unsupervised settings.

What could be improved:

- It is not clear what the authors mean by "none" in tables 1, and 2.

- The proposed method is supposed to work well on cloth-changing person re-identification methods. Although long-term person re-id is not the primary research question of this study, it would be interesting (but not necessary) to compare the proposed method with [1-3].

- In lines 200-202, the authors say that it is possible that their proposed contrastive algorithm selects samples of the same ID. I understand that we do not have the labels and this could happen, however, random selection cannot be a wise solution to resolve this problem. Authors can work on this aspect more. One suggestion: How about, using a simple teacher model that gives a hint to help select contrastive samples wisely?

[1] Proceedings of the Asian Conference on Computer Vision (2020), "Long-term cloth-changing person re-identification." 

[2] Image and Vision Computing (2021), "You look so different! Haven’t I seen you a long time ago?"

[3] IEEE International Conference on Advanced Video and Signal Based Surveillance (2022), "Attribute De-biased Vision Transformer (AD-ViT) for Long-Term Person Re-identification"

Author Response

We sincerely thank the reviewer for the comments and suggestions concerning our manuscript, and we have studied the comments and suggestions carefully. Our responses for the reviewers’ questions are as follows.

Q1. It is not clear what the authors mean by "none" in tables 1, and 2.

Response: Thanks for the reviewer’s question. As introduced in Related Works, existing unsupervised person re-ID methods can be divided into two classes: the unsupervised domain adaptation methods and the purely unsupervised learning methods. We compare our method with both the unsupervised domain adaptation and purely unsupervised learning methods in our experiments. Since the purely unsupervised leaning methods have no corresponding source domains, we fill “none” in the grid to represent this.

Q2. The proposed method is supposed to work well on cloth-changing person re-identification methods. Although long-term person re-id is not the primary research question of this study, it would be interesting (but not necessary) to compare the proposed method with [1-3].

Response: Thanks very much for the reviewer’s suggestion. We have compared our method with [2] on LTCC dataset since we only obtained the code of [2] from Github. The comparison results show that our method performs worse than [2] on LTCC. We think it is reasonable since our model is trained on short-term setting and our model is unsupervised.

Q3. In lines 200-202, the authors say that it is possible that their proposed contrastive algorithm selects samples of the same ID. I understand that we do not have the labels and this could happen, however, random selection cannot be a wise solution to resolve this problem. Authors can work on this aspect more. One suggestion: How about, using a simple teacher model that gives a hint to help select contrastive samples wisely?

Response: Thanks very much for the reviewer’s suggestion. Yes, during our experiments, we have realized that our random selection strategy may select difficult samples having the same person IDs with the query images, and thus bring errors. The reviewer’s suggestion has offered us a very promising future research direction. However, we did not found the suitable teacher model to study during our investigation. Therefore, we expect to train a difficult sample selection model in a self-supervised way and use this model to determine the difficult sample used at each iteration. The model has not been successfully trained yet, but we believe that the trained model can help us get better person re-ID performance.

Reviewer 2 Report

1.     The abstract and conclusion need to be improved. The abstract must be a concise yet comprehensive reflection of what is in your paper. Please modify the abstract according to “motivation, description, results and conclusion” parts. I suggest extending the conclusions section to focus on the results you get, the method you propose, and their significance.

2.     What is the motivation of the proposed method? The details of motivation and innovations are important for potential readers and journals. Please add this detailed description in the last paragraph in section I. Please modify the paragraph according to "For this paper, the main contributions are as follows: (1) ......" to Section I. Please give the details of motivations. In Section 1, I suggest the authors can amend your contributions of manuscript in the last of Section 1.

3.     The description of manuscript is very important for potential reader and other researchers. I encourage the authors to have their manuscript proof-edited by a native English speaker to enhance the level of paper presentation. There are some occasional grammatical problems within the text. It may need the attention of someone fluent in English language to enhance the readability.

4.     The introduction section of the paper needs to revise according to the timeline of technology development. Please update references with recent paper in CVPR, ICCV, ECCV et al and Elsevier, Springer. In your section 1 and section 2, I suggest the authors amend several related literatures and corresponding references in recent years. For example: FFTI: Image Inpainting Algorithm via Features Fusion and Two-Steps Inpainting (Journal of Visual Communication and Image Representation, https://doi.org/10.1016/j.jvcir.2023.103776).

5.     Please give the details of proposed method for proposed model. I suggest the authors amend the calculation of your size of proposed method and the details is important for proposed method.

6.     The content of experiments needs to amend related experiments to compare related SOTA in recent three years. I recommend the authors amend related experimental results of proposed method of SOTA according to the published paper in IEEE, Springer and Elsevier.

7.     However, the manuscript, in its present form, contains several weaknesses. Adequate revisions to the following points should be undertaken in order to justify recommendation for publication.

8.     In the conclusion section, the limitations of this study and suggested improvements of this work should be highlighted.

9.     Provide a critical review of the previous "journal" (not conference) papers in the area and explain the inadequacies of previous approaches.

10.  I suggest the authors revise Section 1 and Section 2. Please revise the content according to the development of timeline. In Figure 5, please amend related depict for some modules.

11.  Please check all parameters in the manuscript and amend some related description of primary parameters. In section 3, please write the proposed algorithm in a proper algorithm/pseudocode format with section 3. Otherwise, it is very hard to follow. Some examples here: https://tex.stackexchange.com/questions/204592/how-to-format-a-pseudocode-algorithm

Author Response

We sincerely thank the reviewer for the comments and suggestions concerning our manuscript, and we have studied the comments and suggestions carefully and made corrections in our manuscript. Our responses for the reviewers’ questions are as follows, and we have made corresponding modifications in our manuscript.

Q1. The abstract and conclusion need to be improved. The abstract must be a concise yet comprehensive reflection of what is in your paper. Please modify the abstract according to “motivation, description, results and conclusion” parts. I suggest extending the conclusions section to focus on the results you get, the method you propose, and their significance.

Response: Thanks for the reviewer’s suggestion. We have modified our Abstract according to the reviewer’s suggestion. The revised Abstract is as follows and we have marked each part:

(Motivation:)State-of-the-art purely unsupervised learning person re-ID methods first cluster all the images into multiple clusters and assign each clustered image a pseudo label based on the cluster result. Then, they construct a memory dictionary that stores all the clustered images, and train the feature extraction network based on this dictionary. All these methods directly discard the unclustered outliers in the clustering process and train the network only based on the clustered images. The unclustered outliers are difficult images with different clothes and pose, low resolution, severe occlusion, and so on, which are frequently appeared on the real-world applications. Therefore, the model trained only on the clustered images will be less robust and unable to handle the difficult images. (Description:)We construct a difficult-image-considered memory dictionary consisting of both the clustered and unclustered images, and design a corresponding contrastive loss by considering both two kinds of images. (Results and Conclusion:)The experimental results show that our difficult-image-considered memory dictionary and contrastive loss can improve the person re-ID performance, which demonstrated the effectiveness of considering the unclustered difficult images in unsupervised person re-ID.

We have also revised the Conclusion based on the reviewer’s suggestion, where we concentrate more on the description, result, and significance of our method. The revised Conclusion is as follows:

We proposed a novel difficult-image-considered cluster-based purely unsupervised learning person re-ID method, where we constructed a difficult-image-considered memory dictionary and contrastive loss by not only considering the easily-clustered common images, but also the difficult images that are hard to cluster. At each iteration, we use the average feature to represent each cluster instance, and randomly sample a difficult image to represent the difficult instance in the memory dictionary, and compute the contrastive loss based on the comparison between the query images with both the two kinds of instances. The experimental results showed that using our proposed memory dictionary and contrastive loss can obviously increase the person re-ID performance, which demonstrated the effectiveness of considering the difficult images during the iterative training process, as well as the effectiveness of our method. We found out that the sampled difficult image sometimes has the same person ID with the query images, which is contrary to our assumption and will therefore bring errors to the final person re-ID result. To tackle this issue, we prefer to use a more effective difficult instance sample strategy, to select a more useful difficult image at each iteration that definitely has different person ID with the query images.

Q2. What is the motivation of the proposed method? The details of motivation and innovations are important for potential readers and journals. Please add this detailed description in the last paragraph in section I. Please modify the paragraph according to "For this paper, the main contributions are as follows: (1) ......" to Section I. Please give the details of motivations. In Section 1, I suggest the authors can amend your contributions of manuscript in the last of Section Response: Thanks for the reviewer’s question. Yes, the motivations and innovations are very important, and we think we have already introduced the motivation of our work clearly in the fourth paragraph of Introduction. The motivation of our work is to consider the difficult images in the cluster-based purely unsupervised person re-ID methods, to make the trained model more robust and suitable for real-world situations. Besides, we have stressed the contribution of our work in the last of Introduction.

Q3. The description of manuscript is very important for potential reader and other researchers. I encourage the authors to have their manuscript proof-edited by a native English speaker to enhance the level of paper presentation. There are some occasional grammatical problems within the text. It may need the attention of someone fluent in English language to enhance the readability.

Response: Thanks for the reviewer’s suggestion. We have carefully check the manuscript and modified the grammatical problems. Besides, we have asked a native English speaker to enhance our paper presentation.

Q4. The introduction section of the paper needs to revise according to the timeline of technology development. Please update references with recent paper in CVPR, ICCV, ECCV et al and Elsevier, Springer. In your section 1 and section 2, I suggest the authors amend several related literatures and corresponding references in recent years. For example: FFTI: Image Inpainting Algorithm via Features Fusion and Two-Steps Inpainting (Journal of Visual Communication and Image Representation, https://doi.org/10.1016/j.jvcir.2023.103776).

Response: Thanks for the reviewer’s suggestion. We have increased some related works in recent years in Related Works, including the work of “FFTI: Image Inpainting Algorithm via Features Fusion and Two-Steps Inpainting”. At present, there are many recent papers in CVPR, ICCV, ECCV are introduced in the Related Works.

Q5. Please give the details of proposed method for proposed model. I suggest the authors amend the calculation of your size of proposed method and the details is important for proposed method. The content of experiments needs to amend related experiments to compare related SOTA in recent three years. I recommend the authors amend related experimental results of proposed method of SOTA according to the published paper in IEEE, Springer and Elsevier.

Response: Thanks for the reviewer’s suggestion. Most methods that we compared are the related SOTA in recent three years, and are from the published paper in IEEE.

Q6. However, the manuscript, in its present form, contains several weaknesses. Adequate revisions to the following points should be undertaken in order to justify recommendation for publication. In the conclusion section, the limitations of this study and suggested improvements of this work should be highlighted.

Response: Thanks for the reviewer’s suggestion. We have analyzed the limitations of our work, and proposed a promising future research direction according to the limitation. The increased sentences in the Conclusion are as follows:

We found out that the sampled difficult image sometimes has the same person ID with the query images, which is contrary to our assumption and will therefore bring errors to the final person re-ID result. To tackle this issue, we prefer to use a more effective difficult instance sample strategy, to select a more useful difficult image at each iteration that definitely has different person ID with the query images.

Q7. Provide a critical review of the previous "journal" (not conference) papers in the area and explain the inadequacies of previous approaches.

Response: Thanks for the reviewer’s suggestion. We have increased some journal papers in the related work, and we have explained the inadequacies of each kind of unsupervised person re-ID methods at the end of the second and third paragraphs in Related Works.

Q8. I suggest the authors revise Section 1 and Section 2. Please revise the content according to the development of timeline. In Figure 5, please amend related depict for some modules.

Response: Thanks for the reviewer’s suggestion. We wrote Introduction and Related Works mainly according to the classification of existing unsupervised person re-ID methods. We think it would be clearer to arrange the content by classification than by timeline, since the difference between our method and other kind of purely unsupervised person re-ID methods, as well as other domain adaptation unsupervised methods can be exhibited clearly in this way. In this case, the motivation and highlights of our method can be introduced more clearly. In addition, there is no Figure 5 in our manuscript.

Q9. Please check all parameters in the manuscript and amend some related description of primary parameters. In section 3, please write the proposed algorithm in a proper algorithm/pseudocode format with section 3. Otherwise, it is very hard to follow. Some examples here: https://tex.stackexchange.com/questions/204592/how-to-format-a-pseudocode-algorithm

Response: Thanks very much for the reviewer’s suggestion. We introduced each parameter carefully when it first appears, and we have checked all the parameters in the manuscript. We have modified Algorithm 1 according to the example that the reviewer offered.

Round 2

Reviewer 2 Report

I have no more new comments.